# The impact of top management team tenure heterogeneity on innovation efficiency of declining firms

Xunjiang Huang[1]*, Qilin Gao[1], Deng Wang[1,2]

1 School of Business Administration, Northeastern University, Shenyang, China, 2 Wuhan Bohong Construction Group Co., Ltd, Wuhan, China

* xjhuang@mail.neu.edu.cn

**Data Availability Statement:** All relevant data are within the manuscript and its Supporting Information files.

**Funding:** This research was funded by the National Office for Philosophy and Social Sciences (CN)

## Abstract

Most firms will experience a decline in their development process. The contraction in demand and the impact of COVID-19 have exacerbated a firm's performance. Under the dilemma of resources reduction and recovery, the declining firm pays more attention to the efficient utilization of the diminishing innovation resources. Based on upper echelon theory and principal-agent theory, this study investigates the impacts of top management team (TMT) tenure heterogeneity and ownership concentration on innovation efficiency. The sample consists of 534 firm observations after PSM nearest-neighbour matching, sourced from A-share listed manufacturing firms in Shanghai and Shenzhen from 2015 to 2019. Innovation efficiency of declining firms is measured using the Malmquist Index method. The fixed-effects (FE) model, PSM-DID model, and stepwise regression are employed to test our hypotheses. The main findings conclude that TMT tenure heterogeneity improves innovation efficiency, and the effect in declining firms is stronger compared to normal firm. Moreover, the concentrated ownership structure inhibits this positive effect because of the excessive tight control over TMT, and this inhibitory effect is stronger in declining firms than normal firms. The robustness checks of alternative variables and alternative regression model, and the addressing of endogenous problem, further support these findings. Efficiency improvement is crucial for the recovery of declining firm. The introduction of an efficiency perspective bridges the gaps in the existing literature. This study contributes to the literature on upper echelon theory and principal-agent theory by integrating them into the context of declining firms. The continuous interaction between the replacement TMT members and ownership restructuring shapes the dynamic capability of declining firms, contributing to the dynamic capability literature. The findings also provide practical guidelines for declining firms, such as replacing top management members and diluting equity, to achieve recovery. It is noted that an excessive focus on efficiency can also lead to neglecting thorough strategic change.

(Grant No. 20BGL044). The funders had no role in study design, data collection and analysis, decision to publish, or preparation of the manuscript.

**Competing interests:** The authors have declared that no competing interests exist.

## 1. Introduction

Organizational decline refers to the continuous deterioration of performance or the continuous erosion of resource base in a specific period [1, 2]. A firm in declining will probably survive, recovery or even die in the future. Organizational aging, transformation and upgrading, rising costs, and demand contraction have worsened the survival crisis of firms, and more and more firms are facing decline [3]. Most, if not all, firms will face decline at a certain stage in their development process [4]. The behavioral theory and prospect theory suggest that organizational decline drives a firm to engage in innovation activities [5, 6]. Innovation activities play an increasingly important role in enhancing financial performance and the core competitive edges [7, 8]. Although product and process innovation can promote the turnaround of declining firms, it also accelerates the consumption of critical resources, potentially dragging them into a downward spiral [9]. The decline undermines the resource base of a firm, reducing the resources available for investment in innovation activities [4]. Faced with a deteriorating resource base, declining firms focus more on the efficient utilization of innovation resources, which is referred to as innovation efficiency. Innovation efficiency refers to the transformation efficiency from the input of innovation resources to creative output, and it is a key factor in turning around a declining firm by maximizing the innovation output [10]. The improvement of innovation efficiency not only increases the innovation output and economic benefits but also saves the limited innovation resources and provides resource support for other recovery activities. The interpretation of the top management team (TMT) regarding the causes of decline shapes the recovery behavior of the firm [11]. The R&D activities are determined by the innovation decisions made by the TMT [7, 12]. Their cognitive and values traits influence the choice of innovation activities for firms, which are key to affecting innovation efficiency and ultimately turning around declining firms [4, 13]. With the increasing customization and diversification of consumer demand, today's firms generally face the challenge of continuously providing innovative product and service portfolios [13]). Moreover, as the complexity of products increases, the heterogeneity of knowledge required for innovation is increasing. The various dimensions of TMT diversity can exert different roles on innovation performance [12]. However, there are still few literatures on the impact of top management team (TMT) heterogeneity on innovation efficiency [12], and even fewer studies on the impact of TMT heterogeneity on recovery through the improvement of innovation efficiency.

TMT is responsible for firm's competitiveness and survival, and the extant literature suggests that a poor TMT is one of the main causes of firm crises. The crisis indicates managerial incompetence, which in turn increases the likelihood of replacing TMT members [8]. The top management change is an important prerequisite for turnaround success [14]. Some diligent senior managers are mistakenly seen as scapegoats for decline [4]. And decline damages their reputation, reduces their compensation, and increases their work pressure [5]. Numerous executives voluntarily choose to "jump ship" [14]. Therefore, the greater the crisis, the higher the likelihood of executive turnover [15–17]. Executive turnover is the prerequisite for the successful turnaround of firm performance [18–20], as it increases TMT tenure heterogeneity in declining firms. Tenure represents valuable experience and knowledge [21], and the extant literature has recognized the impact of TMT tenure heterogeneity on innovation [22]. However, the findings are inconsistent and even contradictory, with a greater focus on growing firms and, less on declining firms. The continuous decline in performance reduces the shareholder value, especially for major shareholders, which increase conflict between them and managers [8]. Ownership structure has an impact on innovation activities [23]. Javeed et al. [24] argue that ownership concentration has a positive effect on innovation because shareholders pursue profit maximization, while Minetti et al. [25] point out that a dispersed ownership structure is

beneficial for diversifying risks in innovation and promoting innovation. Ownership concentration affects the discretion of TMT in operations through the appointment of senior management members and participation in major decisions. However, less attention has been paid to the monitoring mechanisms of shareholders on TMT's recovery activities [8]. The empirical evidence on decline mostly comes from developed countries such as Europe and the United States, while research on the decline-recovery of firms in rapidly growing economies is relatively scarce. Few studies have explored the impact of tenure heterogeneity on innovation efficiency, and consistent conclusions have not been reached [26]. In light of this, this paper takes TMT tenure heterogeneity as the starting point to explore its impact on the innovation efficiency of declining firms, based on the input-output perspective of innovation. Furthermore, the moderating role of ownership concentration in the relationship between TMT tenure heterogeneity and innovation efficiency is also examined.

This study makes marginal contributions to the literature as follows. First, we contribute to literature on firm turnaround by the introduction of innovation efficiency. Previous literature has focused on innovation as an important driver of organizational turnaround [5, 8]. However, the negative effect of innovation activities on turnaround performance, caused by the uncertainty of innovation results and the consumption of limited resources, has not been explored. Furthermore, most existing literature measures the innovation activities of firms from a single perspective of input or output [24]. Innovation efficiency includes both the input and output dimensions of innovation activities. The improvement of innovation efficiency can effectively mitigate the dilemma of resources reduction and recovery faced by declining firms. Second, this study incorporates both TMT tenure heterogeneity and ownership concentration into the analytical framework and explores the contingent conditions involved while revealing the role of TMT tenure heterogeneity on innovation efficiency, thereby integrating the divergent views on the role of TMT tenure heterogeneity on innovation. This study extends the upper echelon theory to the context of declining firms. The interaction between executive changes and shareholder changes deepens the understanding of recovery behavior. The continuous replacement of TMT members provides novel knowledge and promotes the reconfiguration of the resource base, which improve the firm's dynamic capability [27]. Third, while most extant literature on TMT heterogeneity has focused on growth firms, this paper extends it into declining firms, which responds to the call for more research on the turnaround through innovation [2]. The revealed impact of TMT substitution and equity structure adjustment on innovation efficiency provides a new feasible path for the turnaround of declining firms through innovation.

The remainder of this paper is organized as follows. Section 2 reviews the related literature and presents hypotheses. Section 3 describes the data sources, variable definitions, and the models. Section 4 discusses the results. Section 5 conducts robustness checks of the main findings. Section 6 concludes with the main results.

## 2. Literature review and hypothesis development

### 2.1 Literature reviews

TMT composition shapes the interaction among TMT members and affects their knowledge exchange [28]. The characteristics of TMT exert great influence on organizational performance outcomes [29]. TMT diversity makes a firm to pay more attention the innovation field [13]. Upper echelon theory argues that executives with different tenures play different roles in the process of innovation efficiency [30]. Executive tenure is often viewed as the life cycle of managers, influencing their cognitive and learning abilities [31, 32]. Executives with longer tenure have relatively extensive work experience, and can better understand and be familiar

with the firm. Therefore, they conduct R&D activities more effectively [33–35]. However, in their later careers, executives tend to favor risk-averse activities, choosing to maintain the status quo and exclude innovation [36, 37]. Managers with shorter tenures more openly and actively participate in strategic change to improve organizational performance and gain support and trust [21]. Levesque et al. [38] demonstrated that there is a nonlinear relationship between executive tenure and innovation.

TMT tenure heterogeneity is the degree of tenure diversity among TMT members in a firm. Most existing studies explore the impact of TMT tenure heterogeneity on strategic decisions [28, 39, 40] and financial performance [22, 41] based on the upper echelon theory. Some scholars have also studied the relationship between TMT tenure heterogeneity and innovation. They argue that TMT tenure heterogeneity can enrich sources of information and achieve complementary information among members, improving decision-making [28], thus promoting innovation in a positive way [42, 43]. In contrast, Van Knippenberg and Van Ginkel [44] posit that TMT tenure heterogeneity increases costs and decreases team communication efficiency, thus negatively impact innovation activities.

There is an inherent divergence of interests between shareholders and TMT [8]. The continuous decline in performance infringes upon the interests of shareholders, prompting them to increase their supervision over TMT to protect their interests. Ownership concentration refers to the proportion of shares held by major shareholders, reflecting the distribution of the right to participate in decision-making and determining the supervision quality and actual control on the firm [24]. Due to various theoretical perspectives and samples, the conclusions regarding the relationship between ownership concentration and innovation are inconsistent. Lopez and López-Millán [45] argue that a centralized ownership structure is beneficial for improving shareholder monitoring, and it encourages executives to invest more in innovation for long-term development. However, the negative effect of ownership concentration on innovation has also been verified from the perspective of risk aversion and the entrenchment effect [25, 46, 47]. Deng et al. [48], Chen et al. [49], and Agnihotri and Bhattacharya [50] argue that only moderate ownership concentration can promote the long-term development of firms based on principal-agent theory, and there should be an inverted U-shaped relationship between ownership concentration and innovation. However, some studies have found no significant relationship between ownership concentration and innovation [51, 52].

To turn around the declining firm, shareholders carry out recovery activities such as equity transfer, merger and acquisition, strengthening supervision, and replacing executives [5]. Due to the separation of ownership and control, TMT usually has actual control over a firm, their attribution of decline determines the recovery strategic choices of the firm [11]. Therefore, its decision-making must also be regulated by heterogeneous shareholders, leading to differential performance in recovery. However, there is still less attention given to the supervisory roles of shareholders in strategic choice to recover [8]. And the existing literature mainly focuses on the relationship between ownership concentration, TMT heterogeneity, and financial performance [53], with few paying attention to the effect of ownership concentration on the relationship between TMT heterogeneity and innovation [54].

Although studies have explored the role of TMT heterogeneity in influencing innovation from different perspectives and theories, inclusive conclusions have been reached due to different representations of innovation inputs and outputs, among others [42–44]. And the existing studies have primarily focused on growth firms, with few examining the unique scenario of innovation-driven recovery in declining firms. Unlike growth firms, declining firms face the challenge of resource reduction and rapid recovery, which imposes severe resource constraints on their innovation activities, making efficiency improvement a primary concern. Furthermore, declining firms often experience frequent TMT members replacements due to

pressures from boards, suppliers and other stakeholders [14], leading to increased TMT tenure heterogeneity, which in turn affects firms' innovation activities and innovation efficiency. Shareholders, who have ultimate control over the firm, face higher risks of losses and intensify supervision and control over the TMT through member replacements and direct involvement in decision-making process, particularly regarding innovation activities. This contributes to the long-term growth of declining firms. Considering the impact of TMT replacement and ownership adjustment on innovation activities, this paper explores the effect of TMT tenure heterogeneity on the innovation efficiency of declining firms, as well as its contingency conditions. The goal is to uncover the "black-box" through which TMT tenure heterogeneity influences the recovery of declining firms.

## 2.2 Research hypothesis

The expertise and social network resources of TMT members with longer tenure can help firms obtain resources more cheaply and precisely, reducing the cost of innovation [55, 56]. The longer the tenure, the more deeply embedded TMT members become in organizational routines and processes, smoothing the innovation process and accelerating product iteration and updating [57]. Executives who have been with the same firm for a longer period have a better understanding of the firm and the industry environment, enabling them to evaluate, plan and manage innovation projects more scientifically, improving the efficiency of resource allocation for innovation. However, long-tenure executives may exhibit cognitive inertia, making them more inclined to adhere to existing strategies and resistant to change, thus restraining innovation [58]. As TMT tenure lengthens, the organizational climate within the TMT may become more and more conservative, which is not conducive to the high-risk innovative activities, and then blocks the information exchange and knowledge transfer among members, hindering the improvement of innovation efficiency [59].

On the contrary, new TMT members bring new and heterogeneous knowledge to the declining firm, and the combination of different resources stimulates the generation of new ideas [60]. To gain organizational identification, they actively drive the organizational change, optimize organizational processes, and create new product portfolios through stronger risk-taking and work passion [61–63]. However, new TMT members are also unfamiliar with the status of innovation resources, organizational processes, and development strategies [64]. TMT members with different tenures can formulate the most suitable strategy because they have experienced different development stages of the firm and are more likely to have differentiated interpretations of the current performance status [65]. Each member of the TMT with high heterogeneous tenure has unique capabilities, knowledge, and organizational experiences, promoting the generation of optimal investment plans [26]. They are more enthusiastic about sharing knowledge, shaping a more inclusive organizational climate, and enabling the development of creative innovation solutions [28]. Their ability to conceive more innovative actions is enhanced because of a wider range of stimuli [29]. Focusing innovation activities on innovation fields prevents the waste of resources [13]. TMT members with low heterogeneous tenure often possess the same understanding of things, which makes it difficult to generate novel ideas and inhibits innovation. The greater the TMT tenure heterogeneity, the larger the pool of shared information among executives, making it easier for firms to search for higher-return innovation projects and increasing the possibility of project success [66]. Overall, TMTs with high tenure heterogeneity fully utilizes the advantages of long-term and short-term executives [29], which can maximize the output of innovative resources. Although high TMT tenure heterogeneity also increases conflicts and delays decision-making on innovation, entirely

heterogeneous TMTs in management practice are isolated cases [13]. Accordingly, this study argues that TMT tenure heterogeneity improves innovation efficiency.

For declining firms, TMT members with longer tenure are more familiar with the internal and external environments and the status of its resources. This helps firms to make creative use of resources and objectively diagnose innovation processes, enabling them to make more adaptive adjustments for improving the innovation process. It also facilitates firms in identifying and selecting projects that can quickly reverse the current decline and promotes the development of short-cycle innovation projects. The knowledge of a declining firm and the power accumulated over a longer tenure allow TMT members to adopt more aggressive cost-cutting strategies and reduce unnecessary costs in the innovation process. Faced with internal and external pressures, new TMT members have a lower psychological commitment to the firm's current strategy and prefer radical strategic changes [67]. The performance pressures change the strategic inertia of longer-tenured TMT members, leading the firm to reorient its innovation direction and reshape its innovation processes. The involvement of new TMT members effectively mitigates the myopic behavior, cognitive rigidity, and strategic continuity issues in attributing decline, initiated by longer-tenured executives, stimulating the declining firm to increase its endeavours in innovation by launching offensive strategies, such as the introduction of innovative products. TMT members with different tenures have diversified knowledge and experience, enabling them to analyze the causes of decline and strategic changes in greater depth [68]. Facing severe legitimacy pressures, all TMT members are eager to accelerate the implement of innovation activities and turn the firm around to win the support of shareholders, suppliers and other relevant stakeholders.

Based on the above analysis, TMT tenure heterogeneity will positively promote the improvement of innovation efficiency in both normal and declining firms. However, the TMT tenure heterogeneity has a more significant impact on the innovation efficiency of declining firms than that of normal firms due to greater recovery pressure and dwindling resource. Accordingly, this paper proposes the following hypothesis.

H1: Compared with normal firms, TMT tenure heterogeneity has a stronger positive effect on the innovation efficiency of declining firms.

There are both "support effect" [69] and "tunneling effect" [54, 70] between ownership concentration and innovation. Innovation creates long-term value for major shareholders [24], motivating them to strengthen the supervision of executives and encouraging increased investment in innovation activities continuously [71, 72]. The centralized ownership structure reduces internal political power struggles caused by TMT tenure heterogeneity [73, 74], enabling the TMT to be single-minded about innovation and accelerate the implementation of innovation activities. However, the centralized ownership structure will also lead to a series of "tunneling" behaviors, such as private transfer of firm's resources by major shareholders, which will erode the firm's resource base and obstruct the improvement of innovation efficiency [25, 75, 76]. Under the centralized ownership structure, excessive supervision of the TMT reduces the freedom of TMT decision-making, limiting the discretion of the TMT's. The executives tend to avoid high-return innovation activities with high risk, weakening the positive impact of TMT tenure heterogeneity on innovation efficiency [77]. When ownership structure is decentralized, the second agency problem is alleviated, improving innovation resource utilization efficiency through the release of more innovation resources. However, it easily causes "free-rider" problem due to the weakening of the supervisory functions of major shareholders [78]. The executives reduce high-risk innovation activities for self-interested career development and financial control due to a lack of supervision. Although the supervision of major shareholders can directly promote innovation investment and reduce the free riding phenomenon of minor shareholders, with the replacement of senior management

members, major shareholders have strengthened their supervision over the TMT. For their own self-interest, they force the TMT to undertake excessively high-risk innovation projects. Excessive tight control over the TMT reduces their ability to respond to emergencies and their freedom for experimentation [79]. Therefore, this study suggests that the centralized ownership structure inhibits the positive impact of TMT tenure heterogeneity on innovation efficiency.

TMT will face more decision-making constraints from shareholders in the dilemma of decline when decision-making is concentrated in minority shareholders. Major shareholders increase their occupation of the firm's resources through related party transactions, occupation of funds and guarantees to prevent further loss [79]. Threat-rigidity theory holds that organizational decline increases the anxiety, stress, and internal conflict of TMT members [11]. The declining firm becomes conservative, inflexible, and cautious, and its TMT will reduce innovation investment for their own future career reputation. The increase in conflict and stress has led to a gradual centralization of decision-making, which inhibits the willingness of TMT members to share knowledge and reduces the success of relevant decisions. The centralized ownership structure decreases the search area for optimal solutions, limiting it only to a few shareholders or key stakeholders. The capability of the TMT will be questioned and examined by shareholders due to the continuous fall in financial performance, enhancing supervision of the TMT and reducing the discretion of the TMT [9]. The more concentrated the ownership, the more complex the decision-making process for the TMT with strong heterogeneity. The intervention of large shareholders who are unfamiliar with the current operational status further deteriorates the TMT's decision-making on innovation activities, restraining resource allocation in innovation activities and delaying their implementation. The centralized ownership structure of declining firms exposes large shareholders to greater risk of failure in R&D activities. Large shareholders tend to avoid risky innovative projects and increase investment in projects that can generate short-term cash returns. Dispersed ownership concentration increases minority shareholders' influence over the TMT. Their participation reduces the sensitivity of executives to performance fluctuation and increases their risk tolerance [80]. The competitions for resource between short-term innovation projects and high return long-term project leads to a decrease in innovation efficiency, even though the TMT desires for a turnaround for the declining firm.

Based on the above analysis, in the dilemma of decline, shareholders increase their supervision of the TMT, restricting TMT's discretion in selecting innovation activities and delaying the decision-making process. Thus, the concentrated equity structure of declining firms has a stronger negative moderating effect on the relationship between TMT tenure heterogeneity and innovation efficiency than that of normal firms. Accordingly, the following hypothesis is proposed.

H2: Compared with normal firms, the negative moderating effect of ownership concentration of declining firms is more significant.

## 3. Data and methodology

### 3.1 Data and sample

Since 2012, China has intensified reforms of the scientific and technological system and the supervision of listed firms. Considering the impact of COVID-19 pandemic on firm's innovation activities, the data mainly come from the annual data of Shanghai and Shenzhen A-share listed manufacturing firms from 2015 to 2019. Referring to Teixeira et al. [81], firms that meet the following criteria are defined as declining firms: (1) ROA decreases for at least two consecutive years; (2) ROA is negative for at least three consecutive years. Firms with

missing key indicators such as R&D expenditures and number of patents granted are excluded. The patent data are obtained from the CNRDS database, and the financial data are obtained from the CSMAR and WIND databases. Meanwhile, to ensure the comparability of samples, 534 firm observations are finally selected after PSM nearest-neighbour matching on a 1:1 no-put-back basis, of which 267 are declining firms and 267 are normal firms. The DEA-Malmquist index is calculated using Deap2.1. Finally, 2136 sample observations are obtained from 2016 to 2019.

## 3.2 Variables

**Dependent variable.** Innovation efficiency (*Tfpch*). Compared with the DEA model, the Malmquist Index model can measure innovation efficiency in multiple periods. Our research employs the Malmquist Index to represent innovation efficiency. R&D input is the capital stock of R&D expenditure calculated using the perpetual inventory method and the number of R&D personnel in the firm in the current year. Considering the long patent granting cycle, innovation output is represented by sales revenue and the number of patents granted in the lagged one period.

The R&D capital investment is converted into R&D capital stock using model (1) and model (2):

$$K_{it} = K_{it-1}(1 - \delta_{it}) + I_{it} \tag{1}$$

The capital stock in base period:

$$K_0 = I_0/(g + \delta) \tag{2}$$

where $K_{it}$ is the R&D capital stock of firm $i$ in year $t$. $I_{it}$ is the actual R&D capital investment of firm $i$ in year $t$, deflated using 2015 as the base period. $\delta$ is the depreciation rate, and $g$ is the average growth rate of R&D investment. Based on the extant literature, $\delta$ is assumed to take a value of 15% [82]. The price index for R&D inputs is calculated as: 0.45* fixed-asset investment price index + 0.55*consumer price index. Because some declining firms reduce R&D expenses, $g$ may take a negative value. Therefore, $g$ takes a value of 0 when the true value of $g$ is less than 0 in the calculation of the base period capital stock.

**Independent variable.** TMT tenure heterogeneity (*Ten*). The measures of TMT tenure heterogeneity mainly consist of the Herfindal-Hirschman [83] coefficient and the coefficient of variation [84]. Considering that TMT tenure heterogeneity is a continuous variable, we choose the coefficient of variation to denote TMT tenure heterogeneity, with larger values indicating greater TMT tenure heterogeneity. The TMT members mainly include the chairman, (vice) president, (vice) general manager, assistant to the general manager, secretary of the board of directors, directors, supervisors, functional department directors and other managers [40, 85].

**Moderating variable.** Ownership concentration (*Top*1). The proportion of shares held by the first largest shareholder is adopted to represent the ownership concentration [86].

**Control variables.** Based on the extant literature [87–89], we select the following control variables: enterprise size (*Esize*), TMT size (*Tsize*), board size (*Dsize*), solvency (*Asset*), growth ability (*Tobinq*), operational capability (*Growth*), profitability (*Roa*), managerial ownership (*ES*), governmental subsidy (*Sub*), capital intensity (*CI*), proportion of state-owned shares (*State*), equity balances degree (*BS*), and firm age (*Age*). Nature of ownership (*Nature*), industry (*Industry*) and year (*Year*) are controlled simultaneously.

Table 1 provides the variable definitions.

**Table 1. Variable definition.**

| Variables | | Definition |
|---|---|---|
| Dependent variable | *Tfpch* | DEA-Malmquist Index by firm, 2016–2019 |
| Input variables | *Staff* | Number of R&D personnel |
| | *K* | R&D capital stock computed by perpetual inventory method (at constant 2015 prices) |
| Output variables | *Patent* | Number of patents granted in lagged one year |
| | *OI* | Sales revenue (at constant 2015 prices) |
| Independent variable | *Ten* | Coefficient of variation of TMT tenure |
| Moderator variable | *Top*1 | Shareholding ratio of the largest shareholder |
| Control variables | *Tsize* | Ln (number of TMT + 1) |
| | *Dsize* | Ln (number of board members + 1) |
| | *Esize* | Ln (total assets at the end of the year) |
| | *ES* | Ratio of shares held by executives to total shares |
| | *BS* | Total shareholdings of the second to tenth largest shareholders |
| | *CI* | Ln (net fixed assets per capita) |
| | *State* | Ratio of state-owned shares to total shares |
| | *Asset* | Asset-liability ratio |
| | *Sub* | Ratio of subsidy to sales revenue |
| | *Roa* | Rate of return on equity |
| | *Tobinq* | Tobin's q |
| | *Growth* | Total asset turnover |
| | *Age* | Ln (years since the IPO) |
| | *Nature* | Dummy variable, state-owned firm = 1; private firm = 0 |
| | *Industry* | Dummy variable, 1–7 dummy variables based on 2012 SEC industry classification |

## 3.3 Model specifications

Model (3) is constructed to test the role of TMT tenure heterogeneity on innovation efficiency. TMT tenure heterogeneity can promote innovation efficiency if $\beta_1$ is significantly positive and the coefficient of the declining group is larger than that of the normal group, and TMT tenure heterogeneity has a relatively stronger effect on innovation efficiency of declining firms. Thus, Hypothesis 1 can be supported.

$$Tfpch = \beta_0 + \beta_1 Ten + \gamma Controls + \sum Industry + \sum Year + \varepsilon \qquad (3)$$

Model (4) is constructed by introducing the interaction term of ownership concentration and TMT tenure heterogeneity based on model (3) to further test the contingency effect of ownership concentration. If $\beta_2$ is significantly negative and the absolute value of the declining group is larger than that of the normal group, it indicates that the ownership concentration in declining firms exerts a stronger inhibitory effect on the relationship between TMT tenure heterogeneity and innovation efficiency relative to normal firms.

$$Tfpch = \beta_0 + \beta_1 Ten + \beta_2 Top1 + \beta_3 Ten*Top1 + \gamma Controls + \sum Industry + \sum Year + \varepsilon \quad (4)$$

# 4. Results and discussion

## 4.1 Descriptive statistics and correlation analysis

Table 2 shows the descriptive statistics of all variables. As shown in Table 2, the maximum and minimum values of innovation efficiency are 7.697 and 0.025, respectively, and the standard

**Table 2. Descriptive statistics.**

| Variables | Min | Max | SD | Mean | | Median | | Difference test between the two samples | |
|---|---|---|---|---|---|---|---|---|---|
| | Whole | Whole | Whole | Declining | Normal | Declining | Normal | Mean | Median |
| Tfpch | 0.025 | 7.697 | 0.480 | 1.283 | 1.227 | 1.176 | 1.150 | 0.056*** | 0.026 |
| Ten | 0 | 1.985 | 0.283 | 0.575 | 0.571 | 0.563 | 0.568 | 0.004 | -0.005 |
| Top1 | 0.042 | 0.819 | 0.135 | 0.331 | 0.341 | 0.308 | 0.337 | -0.01* | -0.029*** |
| Esize | 19.997 | 27.468 | 1.104 | 22.254 | 22.214 | 22.092 | 22.142 | 0.04 | -0.05 |
| Tsize | 1.099 | 3.219 | 0.321 | 2.054 | 1.984 | 2.079 | 1.946 | 0.07*** | 0.133*** |
| Dsize | 1.609 | 2.89 | 0.162 | 2.228 | 2.236 | 2.303 | 2.303 | -0.008 | 0 |
| ES | 0 | 0.725 | 0.154 | 0.109 | 0.107 | 0.010 | 0.010 | 0.002 | -0.001 |
| BS | 0.015 | 0.663 | 0.121 | 0.249 | 0.234 | 0.244 | 0.223 | 0.014*** | 0.021*** |
| CI | 9.802 | 15.926 | 0.792 | 12.699 | 12.716 | 12.741 | 12.669 | -0.017 | 0.072* |
| Sub | 0 | 5.552 | 0.294 | 0.023 | 0.181 | 0.008 | 0.118 | -0.158*** | -0.004*** |
| Asset | 0.031 | 2.849 | 0.200 | 0.404 | 0.374 | 0.392 | 0.367 | 0.030*** | 0.025** |
| State | 0 | 0.559 | 0.053 | 0.014 | 0.007 | 0 | 0 | 0.006*** | 0 |
| Roa | -41.502 | 1.593 | 1.101 | -0.072 | 0.048 | 0.051 | 0.718 | -0.12** | -0.021*** |
| Tobinq | 0.745 | 14.318 | 1.212 | 1.934 | 2.078 | 1.616 | 1.710 | -0.144*** | -0.094** |
| Growth | 0.039 | 3.645 | 0.346 | 0.539 | 0.636 | 0.484 | 0.538 | -0.097*** | -0.054*** |
| Age | 5.903 | 9.269 | 0.674 | 8.021 | 8.070 | 8.006 | 8.039 | -0.049* | -0.033 |
| N | 534 | 534 | 534 | 267 | 267 | 267 | 267 | 267 | 267 |

Note

***, **, and * represent statistical significance at the 1%, 5%, and 10% levels, respectively.

deviation is 0.480, indicating a significant difference in innovation efficiency among the sample firms. The maximum and minimum values of ownership concentration for the entire sample are 0.819 and 0.042, respectively, with the mean values for the declining group and the normal group being 0.331 and 0.341 respectively. This suggest that the ownership structure of the sample firms is relatively centralized. There are significant differences in ownership concentration between the two groups, both in terms of mean and median values, with the ownership concentration of the declining group being significantly lower than that of the normal group. This indicates that, in the face of declining, major shareholders of declining firms release part of their equity to acquire more resources for recovery. The mean value of TMT tenure heterogeneity is 0.573, with the maximum and minimum values being 1.985 and 0.000, respectively, and the standard deviation is 0.283. This suggests that a significant difference in TMT tenure heterogeneity among the sample firms. The difference test between the two samples shows that the declining firms pay more attention to the efficiency of innovation activities, thus exhibiting higher innovation efficiency.

Table 3 shows the correlation coefficients of the variables. The absolute values of the correlation coefficients among variables are all less than 0.5, indicating the absence of multicollinearity. The correlation coefficient between TMT tenure heterogeneity and innovation efficiency is 0.12, which is significant at the1% level, indicating a significant positive correlation between TMT tenure heterogeneity and innovation efficiency. The greater the TMT tenure heterogeneity, the higher the firm's innovation efficiency. The mean value of VIF is 1.35, and the VIF values among all variables are less than 3. This further supports that there is no serious problem of multicollinearity.

**Table 3. Correlation coefficient of variables.**

| Variable | 1 | 2 | 3 | 4 | 5 | 6 | 7 | 8 | 9 | 10 | 11 | 12 | 13 | 14 | 15 | 16 |
|---|---|---|---|---|---|---|---|---|---|---|---|---|---|---|---|---|
| 1.Tfpch | 1.35 | | | | | | | | | | | | | | | |
| 2.Ten | 0.101*** | 1.24 | | | | | | | | | | | | | | |
| 3.Top1 | -0.064*** | -0.034 | 1.53 | | | | | | | | | | | | | |
| 4.Esize | -0.051** | 0.182*** | 0.068*** | 2.01 | | | | | | | | | | | | |
| 5.Tsize | 0.004 | 0.254*** | -0.050** | 0.247*** | 1.19 | | | | | | | | | | | |
| 6.Dsize | -0.006 | 0.060*** | -0.072*** | 0.210*** | 0.207*** | 1.12 | | | | | | | | | | |
| 7.ES | 0.004 | -0.166*** | -0.076*** | -0.294*** | -0.081*** | -0.176*** | 1.38 | | | | | | | | | |
| 8.BS | -0.090*** | -0.111*** | -0.439*** | 0.017 | 0.035 | -0.016 | 0.246*** | 1.64 | | | | | | | | |
| 9.CI | 0.069*** | 0.081*** | -0.062*** | 0.262*** | 0.034 | 0.069*** | -0.163*** | -0.075*** | 1.15 | | | | | | | |
| 10.Sub | -0.051** | 0.045** | -0.001 | 0.067*** | -0.003 | 0.018 | -0.043** | -0.023 | 0.026 | 1.02 | | | | | | |
| 11.Asset | 0.021 | 0.147*** | -0.067*** | 0.459*** | 0.123*** | 0.148*** | -0.207*** | -0.080*** | 0.187*** | 0.019 | 1.42 | | | | | |
| 12.State | -0.018 | 0.020 | -0.006 | 0.165*** | 0.043** | 0.069*** | -0.119*** | 0.068*** | 0.032 | -0.028 | 0.076*** | 1.06 | | | | |
| 13.Roa | -0.113*** | -0.052** | 0.036* | 0.002 | 0.018 | 0.013 | 0.036* | 0.03 | -0.049** | -0.002 | -0.174*** | -0.009 | 1.05 | | | |
| 14.Tobinq | 0.104*** | -0.079*** | 0.093*** | -0.346*** | -0.041* | -0.084*** | 0.007 | 0.048** | -0.171*** | -0.019 | -0.297*** | -0.083*** | 0.039* | 1.24 | | |
| 15.Growth | -0.121*** | 0.022 | 0.113*** | 0.177*** | -0.021 | 0.088*** | -0.138*** | -0.055** | -0.108*** | 0.053** | 0.135*** | -0.004 | 0.043** | -0.016 | 1.13 | |
| 16.Age | 0.097*** | 0.360*** | -0.092*** | 0.452*** | 0.075*** | 0.219*** | -0.464*** | -0.274*** | 0.174*** | 0.106*** | 0.287*** | 0.153*** | -0.053** | -0.152*** | 0.199*** | 2.02 |

Note

***, **, and * represent statistical significance at the 1%, 5%, and 10% levels, respectively.

**Table 4. Regression results.**

| Variables | Direct effect | | | Moderating effect | | |
|---|---|---|---|---|---|---|
| | (1) | (2) | (3) | (4) | (5) | (6) |
| *Ten* | 0.156*** | 0.230*** | 0.075* | 0.164*** | 0.230*** | 0.084** |
| *Top*1 | | | | -0.116 | -0.243* | 0.027 |
| *C_Ten*C_Top*1 | | | | -1.214*** | -1.554*** | -0.740*** |
| *Esize* | -0.050*** | -0.063*** | -0.010 | -0.043*** | -0.051*** | -0.010 |
| *Tsize* | 0.019 | 0.005 | 0.031 | 0.010 | -0.008 | 0.027 |
| *Dsize* | -0.093 | -0.141 | 0.017 | -0.080 | -0.137 | 0.030 |
| *ES* | 0.143** | 0.130 | 0.234*** | 0.170** | 0.138 | 0.257*** |
| *BS* | -0.104 | -0.023 | -0.251** | -0.183* | -0.15 | -0.245** |
| *CI* | 0.028** | 0.075*** | -0.031* | 0.026 | 0.073*** | -0.031* |
| *Sub* | 0.035 | 0.169 | 0.062 | 0.032 | 0.200 | 0.056 |
| *Asset* | 0.072 | 0.172** | -0.196*** | 0.057 | 0.134* | -0.184* |
| *State* | -0.439** | -0.464* | -0.376 | -0.446** | -0.514** | -0.377 |
| *Roa* | -0.046*** | -0.025** | -0.436*** | -0.044*** | -0.023** | -0.435*** |
| *Tobinq* | 0.005 | 0.018 | 0.006 | 0.007 | 0.014 | 0.007 |
| *Growth* | -0.174*** | -0.299*** | -0.052* | -0.165*** | $-0.27^{2}$*** | -0.052* |
| *Age* | 0.200*** | 0.238*** | 0.165*** | 0.191*** | 0.215*** | 0.170*** |
| *Nature* | -0.088*** | -0.102** | -0.064** | -0.073*** | -0.074* | -0.059* |
| *Industry* | Control | Control | Control | Control | Control | Control |
| *Year* | Control | Control | Control | Control | Control | Control |
| *_Cons* | 0.940*** | 0.546 | 0.830** | 0.894*** | 0.632 | 0.746** |
| $R^2$ | 0.260 | 0.271 | 0.365 | 0.269 | 0.285 | 0.370 |
| *Adj* $R^2$ | 0.251 | 0.255 | 0.351 | 0.260 | 0.267 | 0.354 |
| *N* | 2136 | 1068 | 1068 | 2136 | 1068 | 1068 |

Note

***, **, and * represent statistical significance at the 1%, 5%, and 10% levels, respectively.

## 4.2 Baseline results

Based on model (3), the specific impact of TMT tenure heterogeneity on innovation efficiency is examined using fixed-effects regression. Columns (1), (2), and (3) in Table 4 present the regression results of the effects of TMT tenure heterogeneity on innovation efficiency for the whole sample, the declining group, and the normal group, respectively. In column (1), the coefficient of TMT tenure heterogeneity on innovation efficiency is 0.156, which is significant at the 1% level, indicating that firms with greater TMT tenure heterogeneity have higher innovation efficiency. In column (2), the coefficient of $\beta_1$ is 0.230, larger than that in column (1), and is significant at the 1% level. However, the coefficient of $\beta_1$ in column (3) is 0.075, smaller than that in column (1) and column (2), and is significant only at the 10% level. This indicates that, compared with normal firms, TMT tenure heterogeneity has a stronger role in improving the innovation efficiency of declining firms. When firms face decline, the introduction of new TMT members can improve their innovation efficiency and accelerate their recovery. Thus, Hypothesis 1 is verified.

Model (4) is employed to test the moderating effect of ownership concentration on the relationship between TMT tenure heterogeneity and innovation efficiency. Columns (4), (5), and (6) in Table 4 present the regression results of the moderating effects for the whole sample, the declining group, and the non-declining group, respectively. In column (4), the coefficient of the interaction term $\beta_3$ between TMT tenure heterogeneity and ownership concentration is

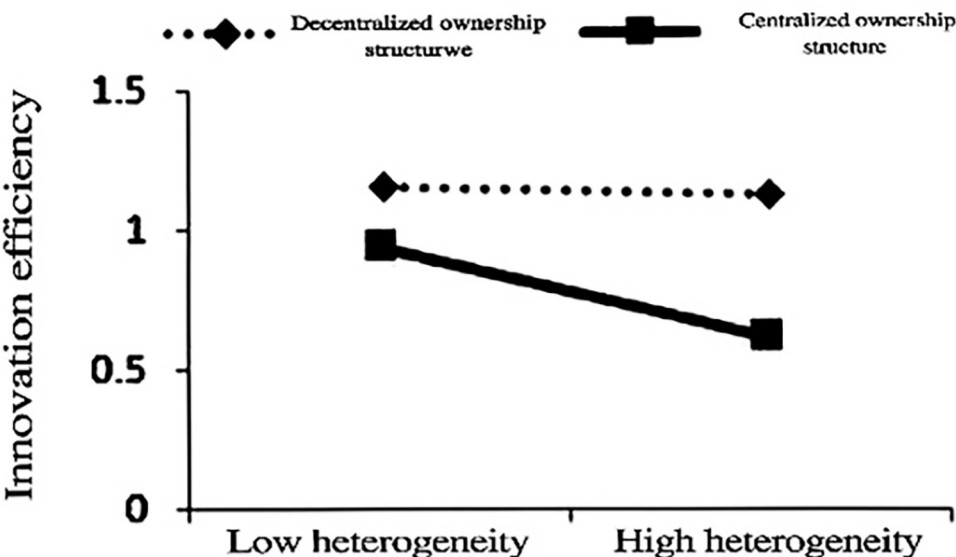

**Fig 1. Moderating effect of the whole sample.**

-1.214, which is significant at the 1% level, indicating that the concentrated ownership structure inhibits the positive effect of TMT tenure heterogeneity on innovation efficiency. Column (5) and (6) show that the coefficient of $\beta_3$ in the declining group is significantly negative at the 1% level, with its absolute value being larger than that of the whole sample. On the other hand, the absolute value of $\beta_3$ in the normal group is smaller than that of the whole sample. This indicates that the ownership concentration of declining firms has a stronger inhibitory effect on the relationship between TMT tenure heterogeneity and innovation efficiency compared to that of normal firms. Therefore, Hypothesis 2 is verified.

The regression coefficients of TMT tenure heterogeneity in columns (4), (5), and (6) in Table 4 further support the Hypothesis 1. Figs 1–3 correspond to the moderating effects of the

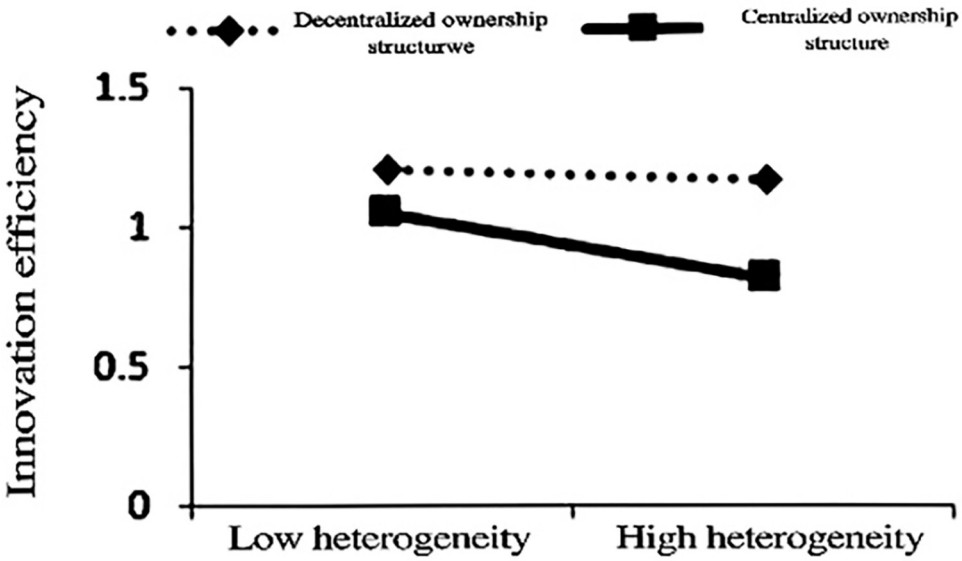

**Fig 2. Moderating effect of the declining group.**

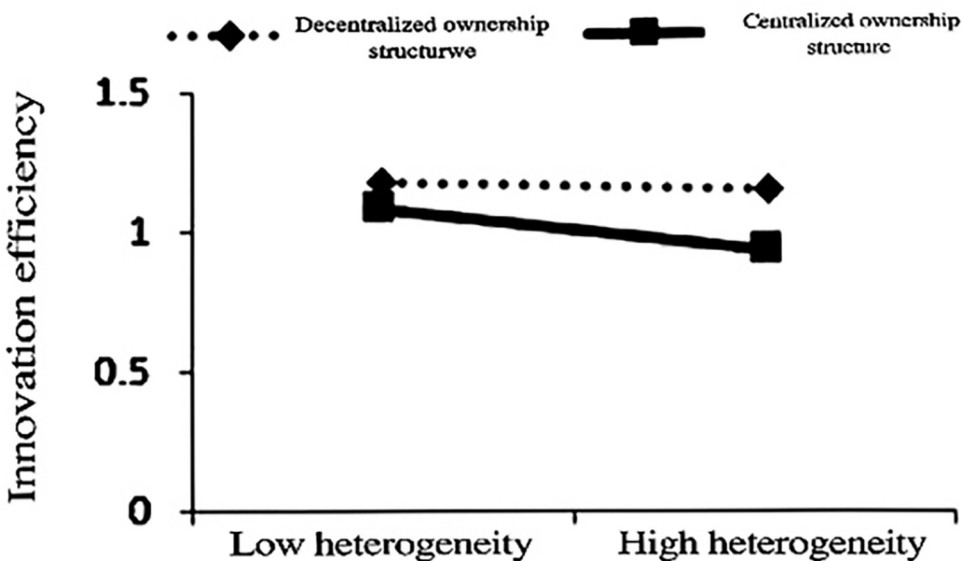

**Fig 3. Moderating effect of the normal group.**

whole sample, the declining group, and the normal group, respectively. It can be seen from the figures that ownership concentration negatively moderates the relationship between TMT tenure heterogeneity and innovation efficiency, with the declining group showing a stronger inhibitory effect. Hypothesis 2 is verified. The moderating effect of ownership concentration has even reversed the positive effect of TMT tenure heterogeneity on innovation efficiency. If a declining firm maintains a highly concentrated ownership structure, the adjustment of TMT will not improve innovation efficiency but rather reduce the efficiency in utilizing innovation resources. Therefore, releasing equity in declining firms can improve the efficiency of resource utilization while obtaining support from new strategic investors with heterogeneous resources.

## 4.3 Endogeneity check

Although we have included as many control variables as possible, the presence of omitted variables poses an endogenous problem. To address this issue, we re-run Model (3) and Model (4) with independent variables lagged one year. The regression results are presented in Table 5. As can be seen from Table 5, the sign and significance level of TMT tenure heterogeneity coefficient $\beta_1$ still support Hypothesis 1. However, the interaction coefficient $\beta_3$ between TMT tenure heterogeneity and ownership concentration in column (5) in Table 5 is not statistically significant, and its absolute value is smaller than that of the whole sample group and the normal group. Hence, Hypothesis 2 does not pass the empirical test when the endogenous problem is accounted for.

There is sample selection bias because the declining and normal sample firms are not randomly selected, which leads to an endogenous problem. Additionally, there is heterogeneity bias due to other unobservable factors in the two sample groups. Therefore, the PSM-DID model is employed to address the endogenous problem. Models (5) and (6) are constructed to verify Hypothesis 1 and Hypothesis 2, respectively.

$$Tfpch = \beta_0 + \beta_1 Lost*Post + \gamma Controls + \sum Industry + \sum Year + \varepsilon \qquad (5)$$

$$Tfpch = \beta_0 + \beta_1 Lost*Post + \beta_2 Lost*Post*Top1 + \gamma Controls + \sum Industry + \sum Year + \varepsilon \quad (6)$$

**Table 5. Regression results with the independent variable lagged one-stage.**

| Variables | Direct effect | | | Moderating effect | | |
|---|---|---|---|---|---|---|
| | (1) | (2) | (3) | (4) | (5) | (6) |
| *Ten* | 0.096** | 0.133* | 0.047 | 0.104*** | 0.137* | 0.053 |
| *Top*1 | | | | -0.149 | -0.364* | 0.047 |
| *C_Ten*C_Top*1 | | | | -0.723*** | -0.596 | -0.715** |
| *Esize* | -0.015 | -0.004 | -0.025 | -0.008 | 0.010 | -0.025 |
| *Tsize* | -0.007 | -0.044 | 0.031 | -0.016 | -0.056 | 0.025 |
| *Dsize* | -0.076 | -0.165 | 0.063 | -0.075 | -0.182 | 0.068 |
| *ES* | 0.048 | 0.020 | 0.105 | 0.062 | 0.008 | 0.127 |
| *BS* | 0.021 | 0.141 | -0.046 | -0.074 | -0.061 | -0.029 |
| *CI* | 0.037** | 0.069*** | -0.004 | 0.034** | 0.063** | -0.004 |
| *Sub* | 0.057 | 0.367 | 0.029 | 0.047 | 0.351 | 0.018 |
| *Asset* | 0.066 | 0.101 | 0.014 | 0.053 | 0.073 | 0.025 |
| *State* | 0.083 | 0.082 | -0.053 | 0.076 | 0.053 | -0.077 |
| *Roa* | -0.037*** | -0.032*** | -0.123 | -0.035*** | -0.030*** | -0.123 |
| *Tobinq* | -0.013 | -0.005 | -0.017 | -0.011 | -0.004 | -0.016 |
| *Growth* | -0.036 | -0.084 | 0.024 | -0.027 | -0.050 | 0.022 |
| *Age* | 0.141*** | 0.162*** | 0.128*** | 0.129*** | 0.131*** | 0.134*** |
| *Nature* | -0.118*** | -0.133** | -0.098*** | -0.106*** | -0.108** | -0.092*** |
| Industry | Control | Control | Control | Control | Control | Control |
| Year | Control | Control | Control | Control | Control | Control |
| *_Cons* | 0.674 | -0.424 | 0.502 | 0.116 | -0.159 | 0.429 |
| $R^2$ | 0.078 | 0.103 | 0.075 | 0.084 | 0.111 | 0.082 |
| *Adj* $R^2$ | 0.065 | 0.077 | 0.048 | 0.069 | 0.082 | 0.053 |
| *N* | 1602 | 801 | 801 | 1602 | 801 | 801 |

Note

***, **, and * represent significance at the 1%, 5%, and 10% levels, respectively.

For firm *i*, if it belongs to the declining group, *Lost* = 1, otherwise, *Lost* = 0. If TMT heterogeneity tenure of firm *i* in the *t*th year is greater than the industry average, *Post* = 1; otherwise, *Post* = 0. *Top*1 represents ownership concentration, *Controls* represent control variables, while controlling for both year and industry effects. The results are presented in columns (1) and (2) in Table 6, where the coefficient $\beta_1$ is significantly positive at the 1% level and the coefficient $\beta_2$ is significantly negative at the 5% level. After considering the endogeneity issue, Hypothesis 1 and Hypothesis 2 are still supported, further conforming the robustness of the regression results.

## 5. Robustness checks

### 5.1 Replace the moderator variable

Ownership concentration is replaced by the proportion of shares held by the top ten shareholders and regressed [86]. The results are shown in Table 6. Columns (3), (4), and (5) of Table 6 show the results after the replacement of the moderator for the whole sample, the declining group, and the normal group, respectively. Apart from the normal group, the interaction term $\beta_3$ between TMT tenure heterogeneity and ownership concentration is no longer significant. The direction and significance level of coefficients for the other two

**Table 6. Endogeneity check and robustness check (PSM-DID model and moderator substitution).**

| Variables | PSM-DID | | Moderator substitution | | |
|---|---|---|---|---|---|
| | (1) | (2) | (3) | (4) | (5) |
| Ten | | | 0.161*** | 0.243*** | 0.075* |
| Top1 | | | -0.127 | -0.244* | 0.028 |
| C_Ten*C_Top1 | | | -0.550** | -0.966** | -0.120 |
| Lost*Post | 0.087*** | 0.210*** | | | |
| Lost*Post*Top1 | | -0.368** | | | |
| Esize | -0.052*** | -0.047*** | -0.043*** | -0.050*** | -0.011 |
| Tsize | 0.034 | 0.031 | 0.012 | -0.007 | 0.031 |
| Dsize | -0.093 | -0.099* | -0.093 | -0.152 | 0.019 |
| ES | 0.145** | 0.142** | 0.149** | 0.116 | 0.236*** |
| BS | -0.116 | -0.169** | -0.066 | 0.080 | -0.267** |
| CI | 0.030** | 0.029** | 0.026* | 0.072*** | -0.030* |
| Sub | 0.061* | 0.062* | 0.033 | 0.184 | 0.061 |
| Asset | 0.062 | 0.052 | 0.061 | 0.144* | -0.191*** |
| State | -0.488*** | -0.506*** | -0.403** | -0.409* | -0.384 |
| Roa | -0.045*** | -0.044*** | -0.045*** | -0.023** | -0.436*** |
| Tobinq | 0.005 | 0.005 | 0.007 | 0.018 | 0.005 |
| Growth | -0.169*** | -0.165*** | -0.169*** | -0.279*** | -0.053* |
| Age | 0.215*** | 0.207*** | 0.193*** | 0.222*** | 0.169*** |
| Nature | -0.092*** | -0.085*** | -0.083*** | -0.094** | -0.064** |
| Industry | Control | Control | Control | Control | Control |
| Year | Control | Control | Control | Control | Control |
| _Cons | 0.856*** | 0.863*** | 0.917*** | 0.591 | 0.803** |
| $R^2$ | 0.258 | 0.260 | 0.262 | 0.278 | 0.366 |
| Adj $R^2$ | 0.250 | 0.251 | 0.253 | 0.260 | 0.350 |
| N | 1602 | 1602 | 2136 | 1068 | 1068 |

Note

***, **, and * represent statistical significance at the 1%, 5%, and 10% levels, respectively.

samples ($\beta_1$, $\beta_3$) remain unchanged, and Hypothesis 1 and Hypothesis 2 are further get supported.

## 5.2 Replace the dependent variable

The innovation efficiency is recalculated by adopting the firm's R&D personnel in the current year and the actual R&D investment in the current year as input variables, and the number of patent applications in the current year and sales revenue in the current year as output variables. The regression results are presented in Table 7. As can be seen from Table 7, the regression coefficients and significance levels of $\beta_1$ and $\beta_3$ in Table 7 still supported the Hypothesis 1 and Hypothesis 2.

## 5.3 Replace the regression model

Since the magnitudes of innovation efficiency values are all greater than 0 and truncated at 0, OLS will produce biased result. Therefore, the Tobit model was employed to verify the two hypotheses, and the results are presented in Table 8. The sign and significance level of the coefficient of TMT tenure heterogeneity, $\beta_1$, and the coefficient of the interaction term, $\beta_3$ still support Hypothesis 1 and Hypothesis 2, indicating that our main findings are still robust.

**Table 7. Robustness check (dependent variable substitution).**

| Variables | Direct effect | | | Moderating effect | | |
|---|---|---|---|---|---|---|
| | (1) | (2) | (3) | (4) | (5) | (6) |
| Ten | 0.181*** | 0.311*** | 0.051 | 0.191*** | 0.312*** | 0.057 |
| Top1 | | | | -0.117 | -0.191 | -0.002 |
| C_Ten*C_Top1 | | | | -1.439*** | -2.215*** | -0.474* |
| Esize | -0.061*** | -0.081*** | -0.015 | -0.053*** | -0.069*** | -0.014 |
| Tsize | 0.033 | 0.008 | 0.054 | 0.024 | -0.007 | 0.051 |
| Dsize | -0.189*** | -0.277** | -0.028 | -0.173** | -0.262** | -0.020 |
| ES | 0.118 | 0.134 | 0.187** | 0.151* | 0.155 | 0.203** |
| BS | -0.084 | -0.049 | -0.209** | -0.166 | -0.155 | -0.217* |
| CI | 0.031** | 0.068*** | -0.018 | 0.030* | 0.067*** | -0.019 |
| Sub | 0.002 | 0.127 | 0.045 | -0.001 | 0.167 | 0.041 |
| Asset | 0.160*** | 0.304*** | -0.190*** | 0.144** | 0.257*** | -0.184*** |
| State | -0.493** | -0.498* | -0.430 | -0.504** | -0.566* | -0.424 |
| Roa | -0.036*** | -0.011 | -0.478*** | -0.034*** | -0.009 | -0.477*** |
| Tobinq | 0.005 | 0.031* | -0.003 | 0.007 | 0.024 | -0.001 |
| Growth | -0.184*** | -0.352*** | -0.043 | -0.174*** | -0.325*** | -0.042 |
| Age | 0.192*** | 0.239*** | 0.152*** | 0.184*** | 0.219*** | 0.153*** |
| Nature | -0.084*** | -0.097* | -0.061** | -0.067** | -0.065 | -0.058* |
| Industry | Control | Control | Control | Control | Control | Control |
| Year | Control | Control | Control | Control | Control | Control |
| _Cons | 1.342*** | 1.23** | 0.958*** | 1.277*** | 1.238** | 0.910** |
| $R^2$ | 0.203 | 0.220 | 0.342 | 0.214 | 0.238 | 0.344 |
| Adj $R^2$ | 0.194 | 0.202 | 0.327 | 0.204 | 0.219 | 0.328 |
| N | 2136 | 1068 | 1068 | 2136 | 1068 | 1068 |

Note

***, **, and * represent statistical significance at the 1%, 5%, and 10% levels, respectively.

# 6. Conclusions and implications

## 6.1 Conclusions

The fierce competition and the pursuit of high-quality development drive firms to pay more attention to the efficient utilization of limited innovation resources. It is even more urgent for declining firms to achieve rapid recovery through the efficient utilization of innovation resources. As the main decision makers, the TMT affects the efficiency of innovation activities. The declining firms adjusts its TMT composition to initiate strategic changes for recovery and increases its heterogeneity level. This study explores the role of TMT tenure heterogeneity on the innovation efficiency of declining firms, and further reveals the boundary conditions of its influential role from the perspective of property rights structure. It is found that TMT tenure heterogeneity can promote the innovation efficiency of declining firms. Organizational decline decreases the firm's legitimacy, and the replacement of TMT members will gain the support of shareholders and creditors. The enrollment of new executives brings new knowledge and resources to the declining firm and reduces resistance to strategic change related to innovation activities. Thus, as the tenure heterogeneity of TMT increases, the innovation efficiency of declining firms improves, and this positive effect is stronger than that in normal firms. It is also supported that the inhibitory role of ownership concentration on the relationship between TMT tenure heterogeneity and innovation efficiency of declining firms is greater than that of

**Table 8. Robustness check (regression model substitution).**

| Variables | Direct effect | | | Moderating effect | | |
|---|---|---|---|---|---|---|
| | **(1)** | **(2)** | **(3)** | **(1)** | **(2)** | **(3)** |
| *Ten* | 0.156*** | 0.230*** | 0.075* | 0.164*** | 0.230*** | 0.084** |
| *Top*1 | | | | -0.116 | -0.243* | 0.027 |
| *C_Ten*C_Top*1 | | | | -1.214*** | -1.554*** | -0.740*** |
| *Esize* | -0.050*** | -0.063*** | -0.010 | -0.043*** | -0.051*** | -0.010 |
| *Tsize* | 0.019 | 0.005 | 0.031 | 0.010 | -0.008 | 0.027 |
| *Dsize* | -0.093 | -0.141 | 0.017 | -0.080 | -0.137 | 0.030 |
| *ES* | 0.143** | 0.130 | 0.234*** | 0.170** | 0.138 | 0.257*** |
| *BS* | -0.104 | -0.023 | -0.251** | -0.183* | -0.158 | -0.245** |
| *CI* | 0.028** | 0.075*** | -0.031* | 0.026* | 0.073*** | -0.031* |
| *Sub* | 0.035 | 0.169 | 0.062 | 0.032 | 0.200 | 0.056 |
| *Asset* | 0.072 | 0.172** | -0.196*** | 0.057 | 0.134* | -0.184*** |
| *State* | -0.439** | -0.464* | -0.376 | -0.446** | -0.514** | -0.377 |
| *Roa* | -0.046*** | -0.025** | -0.436*** | -0.044*** | -0.023** | -0.435*** |
| *Tobinq* | 0.005 | 0.018 | 0.006 | 0.007 | 0.014 | 0.007 |
| *Growth* | -0.174*** | -0.299*** | -0.052* | -0.165*** | -0.272*** | -0.052* |
| *Age* | 0.200*** | 0.238*** | 0.165*** | 0.191*** | 0.215*** | 0.170*** |
| *Nature* | -0.088*** | -0.102** | -0.064** | -0.073*** | -0.074* | -0.059* |
| Industry | Control | Control | Control | Control | Control | Control |
| Year | Control | Control | Control | Control | Control | Control |
| *_Cons* | 0.940*** | 0.546** | 0.830** | 0.894*** | 1.238** | 0.746** |
| *Pseudo R²* | 0.219 | 0.202 | 0.407 | 0.229 | 0.214 | 0.413 |
| *N* | 2136 | 1068 | 1068 | 2136 | 1068 | 1068 |

Note

***, **, and * represent statistical significance at the 1%, 5%, and 10% levels, respectively.

normal firms. The relative concentration of ownership inhibits the innovation discretion of TMT members because it strengthens the intervention of shareholders on the TMT.

## 6.2 Theoretical implications

This study makes four contributions to the literatures. First, the upper echelons theory has been adopted as the predominant theoretical framework to explain the relationship between TMT composition and innovation [28]. Existing literature on upper echelon theory focuses more on growth firms. This study extends it to the context of declining firms facing the dilemma of resource scarcity and decreasing legitimacy. The enrollment of new TMT members introduces new heterogeneous knowledge and resources, helping to gain support from stakeholders. The turnover of old members reduces the resistance to innovative change and contributes to identifying the actual causes of decline more objectively. The substitution of TMT members facilitates firms in searching for more creative solutions to their declining problems and increases TMT tenure heterogeneity, thereby improves innovation efficiency. The application of upper echelon theory helps to illustrate the role of TMT substitution in recovery. Our results provide new insights into the upper echelons theory by revealing the relationship between initiating innovation and top management change in declining firms. Second, this study reveals that the ownership concentration of declining firms has a stronger inhibitory role on the relationship between TMT tenure heterogeneity and innovation efficiency compared to normal firms. Faced with decreasing performance, shareholders increase

supervision on executives to prevent further loss. However, this supervision acts as a disincentive to innovation of executives. The highly concentrated ownership structure exacerbates threat rigidity reactions during declining. This finding suggests that shareholders should release more equity rights or grants TMT more discretion, which differs from the application of principal-agent theory in growth firms and enriches agency theory by revealing its distinct effect on the performance of declining firms. Third, the conventional turnaround strategies are built on the utilization of the firm's current competencies and resources [4]. The continuous replacement of TMT members not only generates more creative problem-solving solutions due to the introduction of new knowledge and resources, facilitating more innovation-oriented strategic change and promoting recovery, but also breaks cognitive rigidity caused by the longer tenure of TMT members, facilitating reconfiguration and reorganization of organizational resources; and organizational adaptation to a changing environment, thus promoting the development of the firm's dynamic capability [27]. Many firms fall into decline not because of a lack of resources and capabilities, but because of their routine inertia and inability to adapt to environmental changes [4]. The planed replacement of TMT fosters the ability to proactively respond to a turbulent environment [13]. Moreover, lower ownership concentration empowers TMT more discretion in strategic changes. With the continuous replacement of TMT members, the ability of declining firms to reconfigure or develop new capabilities is strengthened. Thus, this study contributes to the literature on dynamic capability through the introduction of top management change. Finally, the introduction of the innovation efficiency perspective helps to open the "black box" of the different impacts of TMT heterogeneity on innovation. Most existing studies have analyzed the impact of TMT heterogeneity on innovation from a single perspective of innovation inputs or outputs, and the results differ from each other. Our study argues that the firm pursues both the maximized utilization of limited innovative resources and maximized innovative output due to the performance pressure and resource scarcity in declining, rather than the single optimum of innovation input or output. This bridges the inconsistent findings on the effect of TMT on innovation.

## 6.3 Practical implications

Our study also offers the following practical implications. First, new TMT members bring heterogeneous knowledge and resources, and their desire to change the status quo can effectively overcome organizational inertia in declining firms. The original TMT members are familiar with the operating environment, business processes, etc. The complementary knowledge and abilities of TMT members with different tenure deepen the understanding of the causes of declining and help initiate more realistic strategic changes. The mutual stimulation of new and existing knowledge within the TMT stimulates the creative usage of existing resources and improves innovation efficiency. Therefore, declining firms should pay more attention to the turnover of TMT members, actively introduce new members with creative enthusiasm and an innovation spirit, and retain members who are familiar with the business state and have rich management experience.

Second, as declining firm's shareholders face potential losses, they strengthen their monitoring and control over executives. However, this change may trigger the firm's threat rigidity response and further into downward spirals. Chinese listed firms have a high degree of ownership concentration [23], when the firm faces the dilemma of transformation and upgrading, declining firm's shareholders can proactively introduce new investors to join through equity investment to enrich the firm's resource base. Additionally, giving TMT members the discretion to operate and innovate can accelerate the implement of innovation strategy.

Finally, compared with declining firms, TMT tenure heterogeneity and ownership concentration have less impact on innovation efficiency of normal firms, but it is undeniable that the

reasonable replacement of TMT members can promote innovation efficiency. At the same time, the ownership concentration also has a strong negative inhibitory effect on the innovation efficiency of normal firms. Normal firms should also pay attention to the reasonable dilution of large shareholders' equity to maintain a certain level of resource heterogeneity, stimulate the innovation vitality of executives, and improve innovation efficiency.

The replacement of TMT members can help improve innovation efficiency, but too many turnovers can also lead to chaos and panic. Especially when the new members come from outside the firm, it is difficult for them to know the talent, technology, competence, and other resources required for the successful recovery of the firms [90]. The new members induce conflicts among TMT members, which undermines the effectiveness of turnaround efforts [14]. Declining firms should maintain top management stability in support functions to assist managers in focusing on changes in primary function. Similarly, excessive dispersion of ownership also leads to difficulties in achieving decisions, resulting in delayed implementation of innovation recovery measures. It should be noted that the decline is usually caused by a series of factors. This study only focuses on the roles of TMT tenure heterogeneity and ownership concentration on innovation efficiency. Excessive focus on efficiency cannot drive overall strategic change, making it difficult to enhance its competitiveness [4]. Declining firms should integrate the strategic measures, such as new product introductions, organizational structure change [5], R&D internationalization [8], and digital reorientation [4], with retrenchment activities, such as the divestment, layoffs, and cost reduction, to improve their turnaround performance [5].

### 6.4 Limitations and future research

Our study has several limitations that highlight potential areas for future research. First, TMT heterogeneity has different dimensional compositions [12], our study only analyzes the impact of TMT tenure heterogeneity on innovation efficiency. Future studies can explore the impact of TMT heterogeneity on innovation efficiency in terms of educational background, employment experience, or values. Second, our study only examines the moderating effect of ownership concentration on the relationship between TMT tenure heterogeneity and innovation efficiency. Further studies can analyze the influence of board independence, management power concentration, and other factors to reveal more contingency factors influencing the relationship between TMT tenure heterogeneity and innovation efficiency. Finally, the sample used in this study comes from Chinese manufacturing enterprises listed in Shanghai and Shenzhen A shares from 2015 to 2019. The sample firms mainly come from China, an emerging country, which restricts the generalizability of our findings. More firms in other industries or regions should be included to extend the applicability of the findings. Additionally, more attention should be paid to startups or growth firms experiencing temporary decline, as this study cannot provide suggestions for their turnaround.

## Supporting information

**S1 File.**
(XLSX)

## Acknowledgments

We would like to thank the anonymous reviewers who provided us valuable and constructive comments on the manuscript.

## Author Contributions

**Conceptualization:** Xunjiang Huang.

**Data curation:** Qilin Gao, Deng Wang.

**Formal analysis:** Qilin Gao, Deng Wang.

**Funding acquisition:** Xunjiang Huang.

**Methodology:** Qilin Gao.

**Validation:** Xunjiang Huang, Deng Wang.

**Writing – original draft:** Deng Wang.

**Writing – review & editing:** Xunjiang Huang.

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
