## [Decision Letter · Decision Letter 0]

24 May 2024

PONE-D-24-08354The impact of top management team tenure heterogeneity on innovation efficiency of declining firmsPLOS ONE

Dear Dr. Huang,

Thank you for submitting your manuscript to PLOS ONE. After careful consideration, we feel that it has merit but does not fully meet PLOS ONE’s publication criteria as it currently stands. Therefore, we invite you to submit a revised version of the manuscript that addresses the points raised during the review process.

We look forward to receiving your revised manuscript.

Kind regards,

Ahmed Elamer

Academic Editor

PLOS ONE

Journal Requirements:

This study was funded by the National Office for Philosophy and Social Sciences (CN) (Grant No. 20BGL044)

3. Thank you for uploading your study's underlying data set. Unfortunately, the repository you have noted in your Data Availability statement does not qualify as an acceptable data repository according to PLOS's standards.

Additional Editor Comments:

Please consult more literature from our journal and some of the following:

Guo, B., Pang, X., & Li, W. (2018). The role of top management team diversity in shaping the performance of business model innovation: a threshold effect. Technology Analysis & Strategic Management, 30(2), 241-253.

Hambrick, D. C., Cho, T. S., & Chen, M. J. (1996). The influence of top management team heterogeneity on firms' competitive moves. Administrative science quarterly, 659-684.

Bantel, K. A., & Jackson, S. E. (1989). Top management and innovations in banking: Does the composition of the top team make a difference?. Strategic management journal, 10(S1), 107-124.

Ullah, F., Jiang, P., Elamer, A. A., & Owusu, A. (2022). Environmental performance and corporate innovation in China: The moderating impact of firm ownership. Technological Forecasting and Social Change, 184, 121990.

Ullah, F., Jiang, P., & Elamer, A. A. (2024). Revolutionizing green business: The power of academic directors in accelerating eco‐innovation and sustainable transformation in China. Business Strategy and the Environment.

Ullah, F., Jiang, P., Mu, W., & Elamer, A. A. (2023). Rookie directors and corporate innovation: evidence from Chinese listed firms. Applied Economics Letters, 1-4.

Reviewers' comments:

Reviewer's Responses to Questions

**Comments to the Author**

1. Is the manuscript technically sound, and do the data support the conclusions?

Reviewer #1: Yes

2. Has the statistical analysis been performed appropriately and rigorously? 

Reviewer #1: Yes

3. Have the authors made all data underlying the findings in their manuscript fully available?

Reviewer #1: Yes

4. Is the manuscript presented in an intelligible fashion and written in standard English?

Reviewer #1: Yes

5. Review Comments to the Author

**Reviewer #1:** Abstract

While the abstract effectively introduces the topic of the study and outlines the key factors under investigation, there is room for improvement in terms of providing more specific details about the methodology and findings. Enhancing the abstract with additional information on the study's contributions and practical implications could further engage readers and provide a more comprehensive overview of the research.

Introduction

The introduction could benefit from providing more context on the specific industry or region under study to help readers understand the applicability of the findings. Including information about the industry or industries represented in the sample could enhance the relevance of the study to practitioners in those sectors.

The transition from the general discussion of declining firms to the specific research objectives related to TMT tenure heterogeneity could be smoother.

Literature Review and Hypotheses

If the literature review heavily relies on secondary sources such as literature reviews and meta-analyses, there may be a risk of missing out on recent primary studies or unique perspectives. Incorporating a balance of primary and secondary sources can provide a more comprehensive understanding of the research domain.

The literature review and hypotheses section may be limited in scope, focusing primarily on TMT tenure heterogeneity and ownership concentration. Expanding the discussion to include related concepts or alternative explanations could enrich the theoretical framework and offer a broader perspective on the research topic.

If there are inconsistencies or conflicting findings in the literature, the section should address these discrepancies and provide a nuanced analysis of how the current study aims to contribute to resolving or clarifying such inconsistencies.

The conclusion and implications

The study's conclusions mainly focus on the immediate impact of TMT tenure heterogeneity and ownership concentration on innovation efficiency in declining firms. A weakness lies in the absence of a long-term perspective on how these factors may evolve over time and their sustained effects on the firm's innovation capabilities. Incorporating a longitudinal analysis could offer deeper insights into the dynamics of leadership and ownership structures in driving innovation outcomes.

The study's conclusions and implications are drawn from a specific context (A-share listed manufacturing firms in Shanghai and Shenzhen from 2015 to 2019). The lack of generalizability to firms in other industries or regions limits the broader applicability of the findings. Including a more diverse sample of firms or conducting comparative analyses across industries could enhance the study's relevance to a wider range of organizations.

While the study offers practical implications for declining firms, there is limited discussion on the potential challenges and barriers that organizations may face in implementing the recommended strategies. Addressing the practical hurdles associated with TMT changes, ownership restructuring, and innovation initiatives could provide a more realistic roadmap for firms seeking to improve their innovation efficiency.

The conclusion and implications section could benefit from exploring alternative solutions or strategies beyond TMT tenure heterogeneity and ownership concentration. Considering additional factors or approaches that could impact innovation efficiency in declining firms would enrich the practical insights offered and provide a more holistic view of the challenges and opportunities for organizational recovery.

6. PLOS authors have the option to publish the peer review history of their article (what does this mean?). If published, this will include your full peer review and any attached files.

Reviewer #1: **Yes: **Hammad Bin Azam Hashmi

---

## [Author Response · Author response to Decision Letter 0]

8 Jun 2024

Reply to the Editor and Reviewers to the Report on PONE-D-24-08354

Title of the paper: The impact of top management team tenure heterogeneity on innovation efficiency of declining firms

Dear Editors,

We thank the editor and reviewers for their valuable comments and suggestions on the manuscript and have edited the manuscript to address their concerns.

Based on a combination of comments and suggestions, we have revised the paper and the changes are marked in yellow. Moreover, the most important changes are presented in this reply.

We believe that the manuscript is suitable for publication in PLOS ONE.

Kind regards,

Dr. Huang

On behalf of all authors.

Reply to Journal Requirements:

Requirement 1: When submitting your revision, we need you to address these additional requirements. Please ensure that your manuscript meets PLOS ONE's style requirements, including those for file naming. 

Reply: We have carefully revised the style of this article to adhere to PLOS ONE's guidelines.

Requirement 2: Thank you for stating the following financial disclosure:

 This study was funded by the National Office for Philosophy and Social Sciences (CN) (Grant No. 20BGL044) Please state what role the funders took in the study. If the funders had no role, please state: "The funders had no role in study design, data collection and analysis, decision to publish, or preparation of the manuscript."

Reply: We have included this amended "Role of Funder" statement in our cover letter.

Requirement 3: Thank you for uploading your study's underlying data set. Unfortunately, the repository you have noted in your Data Availability statement does not qualify as an acceptable data repository according to PLOS's standards.

Reply: All relevant data are within the manuscript and its Supporting Information files (S1.xlsx).

Requirement 4: Please amend your list of authors on the manuscript to ensure that each author is linked to an affiliation. Authors’ affiliations should reflect the institution where the work was done (if authors moved subsequently, you can also list the new affiliation stating “current affiliation:….” as necessary).

Reply: The second author has a new job. Therefore, we are listing the new affiliation as follows: "Current affiliation: Wuhan Bohong Construction Group Co., Ltd, Wuhan, China."

Additional Editor Comments: Please consult more literature from our journal and some of the following:

Reply: Thank you for your valuable suggestions. We have reviewed and consulted the new literature as follows:

Abebe, M. A., Tangpong, C., & Ndofor, H. (2024). Hitting the ‘reset button’: The role of digital reorientation in successful turnarounds. Long Range Planning, 57, 102102.

Bai, G., Zhao, J., Xu, P. (2022). How do executives’ synergistic allocation and organizational slack drive enterprise technological innovation? PLoS ONE, 17(10), e0276022.

Barker III, V. L., & Barr, P. S. (2002). Linking top manager attributions to strategic reorientation in declining firms attempting turnarounds. Journal of Business Research, 55, 963-979.

Caselli, S., Gatti, S., Chiarella, C., Gigante, G., & Negri, G. (2023). Do shareholders really matter for firm performance? Evidence from the ownership characteristics of Italian listed companies. International Review of Financial Analysis, 86, 102544.

Guo, B., Pang, X., & Li, W. (2018). The role of top management team diversity in shaping the performance of business model innovation: a threshold effect. Technology Analysis & Strategic Management, 30(2), 241-253.

Hambrick, D. C., Cho, T. S., & Chen, M. J. (1996). The influence of top management team heterogeneity on firms' competitive moves. Administrative science quarterly, 41(4), 659-684.

Javeed, S. A., Latief, R., Jiang, T., Ong, T. S., & Tang, Y. (2021). How environmental regulations and corporate social responsibility affect the firm innovation with the moderating role of Chief executive officer (CEO) power and ownership concentration? Journal of Cleaner Production, 308, 127212.

O'Kane, C., & Cunningham, J. (2014). Turnaround leadership core tensions during the company turnaround process. European Management Journal, 32, 963-980.

Phuong, T. T., Le, A-T., & Ouyang, P. (2022). Board tenure diversity and investment efficiency: A global analysis. Journal of International Financial Markets, Institutions & Money, 81, 101657.

Ren, G., Zeng, P., & Zhong, X. 2024. The effect of innovation performance shortfall on firms’ trade-offs between exploratory and exploitative innovation: Do corporate governance factors matterʔ. Journal of Engineering and Technology Management, 71: 101801.

Shao, Y., Huang, D., Lv, L., & Yu, J. (2021). The influence of non-family members in top management teams on research and development investment: Evidence from Chinese family firms. PLoS ONE, 16(10), e0258200.

Szumowski, W., & Miśkiewicz, R. (2021). The process of developing dynamic capabilities: The conceptualization attempt and the results of empirical studies. PLoS ONE, 16(4), e0249724.

Talke, K., Salomo, S., & Rost, K. (2010). How top management team diversity affects innovativeness and performance via the strategic choice to focus on innovation fields. Research Policy, 39(7), 907-918.

Tangpong, C., Lehmberg, D., & Li, Z. (2024). CEO replacement, top management vacancy, and the sequence of top management team changes in high technology turnaround companies. Long Range Planning, 57, 102103.

Ullah, F., Jiang, P., Elamer, A. A., & Owusu, A. (2022). Environmental performance and corporate innovation in China: The moderating impact of firm ownership. Technological Forecasting and Social Change, 184, 121990.

Wan, W., Zhou, F., Liu, L., Fang, L., Chen, X. (2021). Ownership structure and R&D: The role of regional governance environment. International Review of Economics and Finance, 72, 45-58.

Zhang, Q., Jiang, C., Liu, B., Chan, K. C. (2024). Small but salient: Minority shareholders’ innovation attention in interactive online platforms and corporate innovation. North American Journal of Economics and Finance, 69, 102038.

Additionally, we have checked for spelling errors, grammatical errors, and any other issues. We trust that the revised manuscript will meet the standards of your journal. Please find below our line-by-line response to the reviewers' comments and questions.

Reply to Reviewer #1：

Thank you very much for your feedback. Our point-by-point responses to your comments/questions are provided below.

Comment 1: Abstract

While the abstract effectively introduces the topic of the study and outlines the key factors under investigation, there is room for improvement in terms of providing more specific details about the methodology and findings. Enhancing the abstract with additional information on the study's contributions and practical implications could further engage readers and provide a more comprehensive overview of the research.

Reply: Thank you for your helpful suggestion. We have revised the abstract to include more specific details about the methodology and findings. The updated abstract is as follows:

Most firms will experience a decline in their development process. The contraction in demand and the impact of COVID-19 have exacerbated a firm’s performance. Under the dilemma of resources reduction and recovery, the declining firm pays more attention to the efficient utilization of the diminishing innovation resources. Based on upper echelon theory and principal-agent theory, this study investigates the impacts of top management team (TMT) tenure heterogeneity and ownership concentration on innovation efficiency. The sample consists of 534 firm observations after PSM nearest-neighbour matching, sourced from A-share listed manufacturing firms in Shanghai and Shenzhen from 2015 to 2019. Innovation efficiency of declining firms is measured using the Malmquist Index method. The fixed-effects (FE) model, PSM-DID model, and stepwise regression are employed to test our hypotheses. The main findings conclude that TMT tenure heterogeneity improves innovation efficiency, and the effect in declining firms is stronger compared to normal firm. Moreover, the concentrated ownership structure inhibits this positive effect because of the excessive tight control over TMT, and this inhibitory effect is stronger in declining firms than normal firms. The robustness checks of alternative variables and alternative regression model, and the addressing of endogenous problem, further support these findings. Efficiency improvement is crucial for the recovery of declining firm. The introduction of an efficiency perspective bridges the gaps in the existing literature. This study contributes to the literature on upper echelon theory and principal-agent theory by integrating them into the context of declining firms. The continuous interaction between the replacement TMT members and ownership restructuring shapes the dynamic capability of declining firms, contributing to the dynamic capability literature. The findings also provide practical guidelines for declining firms, such as replacing top management members and diluting equity, to achieve recovery. It is noted that an excessive focus on efficiency can also lead to neglecting thorough strategic change.

Comment 2: Introduction 

The introduction could benefit from providing more context on the specific industry or region under study to help readers understand the applicability of the findings. Including information about the industry or industries represented in the sample could enhance the relevance of the study to practitioners in those sectors.

The transition from the general discussion of declining firms to the specific research objectives related to TMT tenure heterogeneity could be smoother.

Reply: We appreciate your insightful comment. With the rapid development of China's manufacturing industry, the issue of high-quality and sustainable development of Chinese manufacturing firms is increasingly attracting the attention of scholars. The Chinese manufacturing industry consists of 31 sub-sectors (based on level 2 SIC codes) and has extensive industry coverage. It also shares similar characteristics with other countries (Richard et al., 2019). We aim to examine the effects of TMT composition and ownership structure on innovation efficiency in the context of decline, providing practical guidelines for recovery activities. Our hypothesis development and theoretical framework are based on the existing literature, regardless of industry or region. The samples only from the Chinese context are used to test our theoretical hypotheses, which undoubtedly undermines the generalizability of our findings. However, currently, it is limited by the data we are able to collect. In the future, we will compile more samples from various regions to improve our research.

We have revised the introduction to include a clearer and smoother discussion of the research background, research objectives, and theoretical contributions. The updated introduction is as follows.

1. Introduction

Organizational decline refers to the continuous deterioration of performance or the continuous erosion of resource base in a specific period [1-2]. A firm in declining will probably survive, recovery or even die in the future. Organizational aging, transformation and upgrading, rising costs, and demand contraction have worsened the survival crisis of firms, and more and more firms are facing decline [3]. Most, if not all, firms will face decline at a certain stage in their development process [4]. The behavioral theory and prospect theory suggest that organizational decline drives a firm to engage in innovation activities [5,6]. Innovation activities play an increasingly important role in enhancing financial performance and the core competitive edges [7-8]. Although product and process innovation can promote the turnaround of declining firms, it also accelerates the consumption of critical resources, potentially dragging them into a downward spiral [9]. The decline undermines the resource base of a firm, reducing the resources available for investment in innovation activities [4]. Faced with a deteriorating resource base, declining firms focus more on the efficient utilization of innovation resources, which is referred to as innovation efficiency. Innovation efficiency refers to the transformation efficiency from the input of innovation resources to creative output, and it is a key factor in turning around a declining firm by maximizing the innovation output [10]. The improvement of innovation efficiency not only increases the innovation output and economic benefits but also saves the limited innovation resources and provides resource support for other recovery activities. The interpretation of the top management team (TMT) regarding the causes of decline shapes the recovery behavior of the firm [11]. The R&D activities are determined by the innovation decisions made by the TMT [7,12]. Their cognitive and values traits influence the choice of innovation activities for firms, which are key to affecting innovation efficiency and ultimately turning around declining firms [4,13]. With the increasing customization and diversification of consumer demand, today's firms generally face the challenge of continuously providing innovative product and service portfolios [13]). Moreover, as the complexity of products increases, the heterogeneity of knowledge required for innovation is increasing. The various dimensions of TMT diversity can exert different roles on innovation performance [12]. However, there are still few literatures on the impact of top management team (TMT) heterogeneity on innovation efficiency [12], and even fewer studies on the impact of TMT heterogeneity on recovery through the improvement of innovation efficiency.

TMT is responsible for firm's competitiveness and survival, and the extant literature suggests that a poor TMT is one of the main causes of firm crises. The crisis indicates managerial incompetence, which in turn increases the likelihood of replacing TMT members [8]. The top management change is an important prerequisite for turnaround success [14]. Some diligent senior managers are mistakenly seen as scapegoats for decline [4]. And decline damages their reputation, reduces their compensation, and increases their work pressure [5]. Numerous executives voluntarily choose to “jump ship” [14]. Therefore, the greater the crisis, the higher the likelihood of executive turnover [15-17]. Executive turnover is the prerequisite for the successful turnaround of firm performance [18-20], as it increases TMT tenure heterogeneity in declining firms. Tenure represents valuable experience and knowledge [21], and the extant literature has recognized the impact of TMT tenure heterogeneity on innovation [22]. However, the findings are inconsistent and even contradictory, with a greater focus on growing firms and, less on declining firms. The continuous decline in performance reduces the shareholder value, especially for major shareholders, which increase conflict between them and managers [8]. Ownership structure has an impact on innovation activities [23]. Javeed et al. [24] argue that ownership concentration has a positive effect on innovation because shareholders pursue profit maximization, while Minetti et al. [25] point out that a dispersed ownership structure is beneficial for diversifying risks in innovation and promoting innovation. Ownership concentration affects the discretion of TMT in operations through the appointment of senior management members and participation in major decisions. However, less attention has been paid to the monitoring mechanisms of shareholders on TMT's recovery activities [8]. The empirical evidence on decline mostly comes from developed countries such as Europe and the United States, while research on the decline-recovery of firms in rapidly growing economies is relatively scarce. Few studies have explored the impact of tenure heterogeneity on innovation efficiency, and consistent conclusions have not been reached [26]. In light of this, this paper takes TMT tenure heterogeneity as the starting point to explore its impact on the innovation effici

---

## [Decision Letter · Decision Letter 1]

29 Oct 2024

The impact of top management team tenure heterogeneity on innovation efficiency of declining firms

PONE-D-24-08354R1

Dear Dr. Huang,

We’re pleased to inform you that your manuscript has been judged scientifically suitable for publication and will be formally accepted for publication once it meets all outstanding technical requirements.

Kind regards,

Fawad Ahmed

Academic Editor

PLOS ONE

Additional Editor Comments (optional):

Reviewers' comments:

Reviewer's Responses to Questions

**Comments to the Author**

Reviewer #2: All comments have been addressed

2. Is the manuscript technically sound, and do the data support the conclusions?

Reviewer #2: Yes

3. Has the statistical analysis been performed appropriately and rigorously? 

Reviewer #2: Yes

4. Have the authors made all data underlying the findings in their manuscript fully available?

Reviewer #2: Yes

5. Is the manuscript presented in an intelligible fashion and written in standard English?

Reviewer #2: Yes

6. Review Comments to the Author

Reviewer #2: The manuscript is a well-crafted and thoroughly researched piece of work. The authors have successfully addressed the comments and suggestions raised in the previous draft, demonstrating their commitment to improving the quality and rigor of the manuscript. I commend the authors for their diligent work and the substantial improvements made. Overall, this is a commendable contribution to the field, and I congratulate the authors on their successful efforts.

7. PLOS authors have the option to publish the peer review history of their article (what does this mean?). If published, this will include your full peer review and any attached files.

Reviewer #2: **Yes: **Dr. Amjad Iqbal

---

## [Editor Report · Acceptance letter]

31 Oct 2024

PONE-D-24-08354R1 

PLOS ONE

Dear Dr. Huang, 

I'm pleased to inform you that your manuscript has been deemed suitable for publication in PLOS ONE. Congratulations! Your manuscript is now being handed over to our production team.

Kind regards, 

on behalf of

Dr. Fawad Ahmed 

Academic Editor

PLOS ONE